# COVID-19 Prevalence among Czech Dentists

**DOI:** 10.3390/ijerph182312488

**Published:** 2021-11-27

**Authors:** Jan Schmidt, Vojtech Perina, Jana Treglerova, Nela Pilbauerova, Jakub Suchanek, Roman Smucler

**Affiliations:** 1Department of Dentistry, Charles University, Faculty of Medicine in Hradec Kralove and University Hospital Hradec Kralove, 500 05 Hradec Kralove, Czech Republic; Jan.Schmidt@lfhk.cuni.cz (J.S.); Nela.Pilbauerova@lfhk.cuni.cz (N.P.); SuchanekJ@lfhk.cuni.cz (J.S.); 2Department of Oral and Maxillofacial Surgery, Masaryk University, Faculty of Medicine and University Hospital Brno, 625 00 Brno, Czech Republic; treglerova.jana@fnbrno.cz; 3Czech Dental Chamber, Slavojova 270/22, 128 00 Prague, Czech Republic; Smucler@dent.cz

**Keywords:** COVID-19, SARS-CoV-2, prevalence, dentistry, pandemic, dentist, occupational health, infection

## Abstract

This work evaluates the prevalence of coronavirus disease (COVID-19), a viral infection caused by severe acute respiratory syndrome coronavirus 2 (SARS-CoV-2), among members of the Czech Dental Chamber. The assessment was based on an online questionnaire filled out by 2716 participants, representing 24.3% of all chamber members. Overall, 25.4% of the participants admitted they were diagnosed with COVID-19 by 30 June 2021, with no statistical differences between the sexes. While in the age groups under 50 the reported prevalence was around 30%, with increasing age, it gradually decreased to 15.2% in the group over 70 years. The work environment was identified as a place of contagion by 38.4% of the respondents. The total COVID-19 PCR-verified positivity was 13.9%, revealing a statistically lower prevalence (*p* = 0.0180) compared with the Czech general population, in which the COVID-19 PCR-verified positivity was ~15.6% (fourth highest rank in the world). The total infection–hospitalization ratio (IHR) was 2.8%, and the median age group of hospitalized individuals was 60–70 years. For respondents older than 60 years, the IHR was 8.7%, and for those under 40 years, it was 0%. Of the respondents, 37.7% admitted that another team member was diagnosed with COVID-19, of which the most frequently mentioned profession was a nurse/dental assistant (81.2%). The results indicate that although the dentist profession is associated with a high occupational risk of SARS-CoV-2 infection, well-chosen antiepidemic measures adopted by dental professionals may outweigh it.

## 1. Introduction

Coronavirus disease (COVID-19) is a viral infection caused by the newly isolated severe acute respiratory syndrome coronavirus 2 (SARS-CoV-2). The standard clinical features are of a wide flulike spectrum, including fatigue, taste and smell loss, cough, headache, or fever. However, in some patients, it can lead to a more severe form, including breathing difficulties, respiratory failure, or acute inflammatory response, which could be fatal [1,2]. The rapid spread of SARS-CoV-2 is mainly due to the type of its transmission from person to person via respiratory droplets or mucosal contact or less often by contact with fomites [3,4,5]. The first official case of SARS-CoV-2 was reported in Wuhan City, Hubei Province, China, in December 2019 [6]. Due to its global spread, it soon became a worldwide health threat broadly affecting human society and leading the World Health Organization to classify COVID-19 as a pandemic disease as of 11 March 2020 [7].

The first cases of COVID-19 were recorded in the Czech Republic at the beginning of March 2020. The Czech government quickly issued a number of antiepidemic measures, which made the virus spread very limited. At the end of August 2020, the cumulative numbers of COVID-19 PCR-verified cases and total deaths per 100,000 people were 230 and 4, respectively [8,9]. However, since September 2020, the number of infected patients has risen sharply. During the autumn of 2020 and the spring of 2021, the Czech Republic was one of the countries most affected by COVID-19. As of the reference period of this study (i.e., 30 June 2021), the Czech Republic had 15,546 cumulatively PCR-verified infected per 100,000 people, which was the fourth highest number in the world [10]. On the same date, the number of total deaths related to COVID-19 per 100,000 people was 283, which was the fourth highest number in the world [11].

The transmission of SARS-CoV-2 is mainly via droplets, and in areas where there is a great fluctuation and accumulation of individuals, the spread of the disease is heightened. This also applies to medical facilities, making healthcare professionals vulnerable to COVID-19, with a special risk for those whose work is associated with mucus and saliva droplets. This is especially true for dental professionals. The dentist’s work is associated with close contact with many people and producing a large amount of aerosol containing the patient’s saliva and mucus droplets. Due to the high speed of dental rotary instruments, the aerosol swirls at a high speed to a distance of several meters from the source. Thus, the work environment of dentists is particularly risky, and dentists are one of the highly vulnerable groups [12].

During the COVID-19 pandemic, general healthcare was suppressed in the Czech Republic. However, a survey performed among members of the Czech Dental Chamber revealed that Czech dentists worked even throughout the pandemic [13]. During the spring of 2020, in the Czech Republic also called the “first wave” of COVID-19, more than 90% of the participating dentists replied that their practices were open. During the period from autumn 2020 to spring 2021, also called the “second wave” of COVID-19, more than 96% of them replied their practices remained open. From those who closed their practices during the period from March 2020 to March 2021, only less than 10% reported that the closure was longer than 4 weeks. The data showed that Czech dentistry remained very operational during the whole pandemic. This approach was rare on a European and global scale [13]. Such conditions make Czech dentists a unique study group to assess the impact of COVID-19 on dental professionals as it minimizes the bias resulting from their workplace.

Based on the combination of these three factors—high national prevalence, a significant risk of infection due to work settings, and high workload during pandemics—Czech dentists form a unique group with a presumption of high COVID-19 prevalence. At the same time, it could be assumed that dentists will be more affected by COVID-19 than the Czech general population due to the work environment. Furthermore, as Czech dentists remained more operative during the pandemic than their counterparts in other countries, it can be assumed that the regional impact of COVID-19 on this professional group was greater. However, these assumptions are hypotheses only and have not yet been addressed in any study.

On the other hand, Czech dentists were aware of these risks, and in order to maintain high operability, they adopted strict antiepidemic measures, such as an anamnestic questionnaire for each patient, regular testing of dental team members, planning a daily schedule to minimize patient accumulation in dental practices, rubber dam use, barrier precautions, minimizing aerosol spread, or establishing dental centers for the treatment of COVID-19-positive patients. These measures were aimed at minimizing the risk of transmission from patients to staff and vice versa, between staff, and between patients. The Czech Dental Chamber was one of the first dental chambers in Europe to issue antiepidemic recommendations for its members, and ordinary members of the chamber were also very proactive in this regard. These thorough measures could significantly reduce the risk of COVID-19 transmission in dental practices. However, so far, there are no data available to confirm this assumption.

To reflect the need to obtain statistically relevant quantifying data, the Czech Dental Chamber decided to conduct a survey among its members, the results of which are presented in this study.

The aim of this work is to assess the impact of COVID-19 on Czech dentists.

## 2. Materials and Methods

### 2.1. Design

This ad hoc, self-administered, cross-sectional, online survey was conducted by the Czech Dental Chamber and filled out by chamber members. All participants were informed about the purpose of the study, and none of them had a patient status. The questionnaire was anonymous; reported data did not include any identifying information that could be used to trace the participants and did not allow any association with the person answering. The participants were not rewarded with any direct benefits for participating in the survey. This study was conducted in accordance with the Declaration of Helsinki.

The presented data were obtained from the answers to 9 questions. Out of these questions, 8 were close-ended, and 1 was semi-close-ended (prefilled close-ended answers along with the option to reply in an open form). The whole questionnaire was in the Czech native language and was designed in collaboration with experts from the chamber, the academic community, and general practitioners.

A description of the questions, including the type and number of answers, is given in Table 1.

### 2.2. Sample

To address the members of the Czech Dental Chamber, invitations for participation in the survey were sent to all 9922 officially registered e-mail addresses in the chamber database. Each address represents one chamber member. The addressees were asked to fill out the questionnaire from 23 June to 4 September 2021. According to the Czech Dental Chamber 2020 Annual Report, the chamber had 11,160 members as of 31 December 2020 [14]. Thus, the survey addressed 88.9% of the chamber members. Membership in the Czech Dental Chamber is compulsory for all dentists working in the Czech Republic.

### 2.3. Sample Size Relevancy

Based on the total number of chamber members, the minimum relevant number of survey participants was set at 372. This quantification was assessed by the online Netquest calculator using Formula (1). For the calculation, a study universe of the members of the Czech Dental Chamber (N = 11,162), a margin of error of 5%, a confidence level of 95%, and a standard heterogeneity of 50% were used. As the sample size of this study (2716 participants) significantly exceeds the minimum required value (*n* = 372), the results are statistically relevant.
(1)$$n\;=\frac{\;N\cdot Z^{2}\cdot p\cdot \left(1-p\right)}{\left(N-1\right)\cdot e^{2}+Z^{2}\cdot p\cdot \left(1-p\right)}$$

Formula (1). Relevant sample size calculation. Sample size calculated (*n*), size of the universe (N), deviation from the mean value (Z), maximum margin of error tolerated (e), expected proportion (p).

### 2.4. Data Collection

The invitation to participate was sent by e-mail to 9922 officially registered e-mail addresses of the chamber members. The e-mail contained a link to an online questionnaire in Google Forms (Google, Mountain View, CA, USA). The compatibility of the questionnaire interface was not limited and included a mobile phone, desktop computer, laptop, or tablet with support for all the most used operating systems. The collected data were stored in the Google Forms cloud database and downloaded after the whole survey was completed.

### 2.5. Statistical Analysis

After the survey was completed, the results of all the questions were downloaded from the Google Forms cloud database. The results of close-ended questions (Q1–6, Q8, and Q9) were analyzed and presented as the percentage of individual answers within all the answers provided. Blank responses were not included in the total number of responses.

Responses to the semi-close-ended question (Q7) were analyzed individually. Each open-ended answer was evaluated independently by two authors (J.S. (Jan Schmidt), V.P.). Results disagreeing between the authors were resolved by a decision of the third author (J.T.). Open responses that were of similar meaning to closed responses were transferred to the appropriate closed response category. The remaining answers were put into new groups according to their meaning. Newly formed groups that exceeded the specified limit in frequency (*n* = 5) were presented as separate answers within the results. Answers that did not exceed this limit were classified in the “Others” category. Results were analyzed and presented as the percentage of individual answers within all answers provided. Blank responses were not included in the total number of responses.

To compare the COVID-19 prevalence between the Czech Dental Chamber members and the Czech general population, it was necessary to use the same methodology. The available COVID-19 prevalence rate within the Czech general population was based on PCR-confirmed cases and did not include cases diagnosed with clinical symptoms. As of the end of this survey, the COVID-19 cumulative cases among the Czech general population was 15,546 per 100,000 people [10]. In order to compare these values with the results of our study, only PCR-verified diagnoses were used.

The data were analyzed using custom Microsoft Office Excel formulas (version 2106 for Windows, Microsoft Corporation, Redmond, WA, USA) and GraphPad Prism (version 8.0.0 for Windows, GraphPad Software, San Diego, CA, USA). Chi-square with test Yates’s correction was used for statistical analysis; * indicates *p* < 0.05.

## 3. Results

### 3.1. Response Rate

A total of 2716 respondents took part in the survey. Based on the 9922 e-mails sent, the response rate was 27.4%, representing 24.3% of all the chamber members (*n* = 11,162) (Figure 1).

### 3.2. Sex Distribution

A total of 2708 respondents stated their sex, and 8 skipped this question. A total of 1871 (68.9%) selected the female option, and 837 (30.8%) selected the male option (Figure 2), which also corresponds to the dominant representation of women among Czech dentists (64.9%) [14].

### 3.3. Age Distribution

A total of 2712 respondents stated their age, and 4 skipped this question. The distribution is illustrated in Figure 3 and approximately corresponds to the age composition of the chamber members [14]. The median age group is 50–60 years.

### 3.4. COVID-19 Prevalence

#### 3.4.1. COVID-19 Prevalence in the Whole Study Population

A total of 2716 respondents replied to this question. No respondent skipped this question. The results are presented in Figure 4. These data reveal that 691 (25.4%) respondents admitted they were diagnosed with COVID-19 by 30 June 2021.

#### 3.4.2. COVID-19 Prevalence Based on Sex

Sex-based COVID-19 prevalence is provided in Figure 5. Detailed data about the answers provided are available in the Appendix A.

#### 3.4.3. COVID-19 Prevalence Based on Age

Age-based COVID-19 prevalence is illustrated in Figure 6. Age- and sex-based COVID-19 prevalence is shown in Figure 7. Detailed data about the answers provided are available in the Appendix A.

### 3.5. COVID-19 Diagnostics

This question was addressed only to the respondents who confirmed they were diagnosed with COVID-19 in Q3 (*n* = 691). A total of 651 (94.2%) respondents reported 1328 answers to this multiple-choice question. The results are presented as a number of answers, percentage of respondents choosing this answer, and frequency of an answer among all answers in Figure 8.

There were a total of 520 respondents who chose to answer either “clinical symptoms” or “taste and smell loss”. An answer containing some type of test was selected by 496 respondents. The intersection of these two groups was 365 respondents. In 76.2% of the respondents, the diagnosis of COVID-19 was confirmed by a test. In 23.7%, it was diagnosed solely on the basis of clinical symptoms. In 57.9%, the diagnosis was confirmed with a PCR test.

### 3.6. Comparison of COVID-19 Prevalence among the General Population in the Czech Republic

As of the end of this survey, the COVID-19 cumulative cases among the Czech general population was 15,546 per 100,000 people [10]. The PCR-verified prevalence within our study is 13.9%. Compared with the PCR-verified positivity in the general population, the difference is statistically significant (*p* = 0.0180) (Figure 9).

### 3.7. Place of Treatment

This question was addressed only to the respondents who confirmed they were infected with COVID-19 in Q3 (*n* = 691). A total of 646 (93.5%) respondents answered this question.

The vast majority of the participants (628, 97.2%) answered that they were being treated in the household. Only 2.8% of COVID-19 cases led to hospitalization (Figure 10). The median age group of those hospitalized was 60–70 years. In the group of respondents older than 60 years, the infection–hospitalization ratio (IHR) was 8.7%. On the other hand, none of the hospitalized were under the age of 40; the IHR under the age of 40 was 0%.

### 3.8. Awareness of Where the Infection Occurred

This question was addressed only to the respondents who confirmed they were diagnosed with COVID-19 in Q3 (*n* = 691). Of them, 650 (94.1%) respondents answered this question. The results are provided in Figure 11.

### 3.9. Environment Where the Infection Occurred

This question was addressed only to the respondents who reported that they knew or suspected where they were infected within Q8 (*n* = 518). Of them, 517 (99.8%) respondents answered this question.

These results show that 199 (38.4%) respondents identified the work environment as a source of infection. Together with the domestic environment (47.0%), these two categories were the dominant source of infection among respondents, jointly responsible for 85.5% of the reported transmission (Figure 12). Detailed data about the answers provided are available in the Appendix A.

### 3.10. Prevalence of COVID-19 among Other Team Members

This question was addressed to all survey participants. Of them, 1683 (62.3%) replied that they were not aware of any other team member who was ill with COVID-19. A total of 1018 (37.7%) respondents admitted that another team member was ill with COVID-19.

Of the respondents who admitted they were diagnosed with COVID-19, 43.3% also reported another team member who was diagnosed as well, and 55.6% reported that no additional team member was diagnosed. Among those respondents who replied they were not diagnosed with COVID-19, 34.7% also reported another team member who was diagnosed with COVID-19, while 65% reported that no additional team member was diagnosed with COVID-19.

### 3.11. Profession Specification among Other Members of the Dental Team Infected with COVID-19

Those who reported an additional team member diagnosed with COVID-19 in the previous question (a total of 1018 participants, 37.7%) were asked to specify the profession of the infected individual. Of them, 990 (97.3%) replied, providing 1124 answers to this multiple-choice question. The results are presented as a number of answers, percentage of respondents choosing this answer, and frequency of an answer among all answers in Figure 13.

## 4. Study Limitations

There was one limitation that the authors had to address when planning this study and that they would like to discuss in this section. This limitation was not accidentally identified during the survey but was known to the authors before the research began. This chapter describes the limitation causes, possible approaches, and the approach by which the authors decided to address it.

The aim of the study is to describe the impact of COVID-19 on chamber members. In order to describe the prevalence of this disease among the respondents, it was necessary to establish diagnostic criteria. The authors considered whether these criteria would include only test-verified infections or whether they would be accepted together with diagnosis based on sole clinical symptoms.

Criteria based exclusively on tests would enhance the validity of the data. However, this method would lead to skewed results, as a large part of the Czech population was not tested and passed COVID-19 without test confirmation. At the time of the pandemic, test sites were overloaded due to the massive community-based virus spreading, and testing was unavailable to many patients. It is also important to note that one of the recommendations of the Ministry of Health of the Czech Republic was that people with COVID-19 should stay at home and be treated at home unless their condition is serious. The aim of this measure was to keep people with symptoms of COVID-19 in isolation and not to spread the infection just because of laboratory verification of the infection. Such a measure was medically correct but led to the real prevalence of COVID-19 among the population being significantly higher than the prevalence confirmed by the test.

We had two options to address this fact in determining the prevalence of COVID-19 among the study participants. One of them was to consider infected only those respondents in which positivity for COVID-19 was confirmed by a test. This option would lead to the acquisition of meticulous solid data but would significantly differ from the real prevalence. The second option was to accept the infection status regardless of the diagnostic method (i.e., both test-verified diagnosis and diagnosis based on clinical symptoms). This option would lead to less solid total data gain but would better correspond to the actual situation. In the end, we decided to obtain data combining the benefits of both of the abovementioned options.

In order to avoid skewing the results, we decided to include in the study both the group with the test-confirmed infection and the group diagnosed on the basis of clinical symptoms. To be able to distinguish these two groups in the results, the respondents were asked to indicate how COVID-19 was diagnosed, including sorting by the test used for diagnosis. Thanks to this procedure, the survey was as inclusive as possible, methodologically reflecting the epidemiological situation in the country and at the same time providing meticulous solid data. We consider this procedure to be appropriate, as it offers as much data as possible, within which it is still possible to sort the results on the basis of preferred criteria, such as test-verified infections.

## 5. Discussion

As there were no relevant quantitative data on the COVID-19 impact on Czech dentists, the Czech Dental Chamber decided to issue a survey among its members addressing their COVID-19 anamnesis. The data from this survey are presented in this study. Compared with studies with a similar focus and methodology, our work is one with the highest nationwide participation rates [15,16].

As mentioned in the Introduction, it was assumed that the prevalence of this disease would be high in this group. This assumption was confirmed as 25.4% of the respondents stated that they were diagnosed with COVID-19. Of the total reported positive cases among the respondents, the data show that the prevalence was 26.4% among females and 23.3% among males. An interesting phenomenon was observed across age groups. While in the age groups under 50 years, the prevalence was around 30%, with increasing age, it gradually decreased. In the group of 50–60 years, it was 24.8%, in the group of 60–70 years 20.7%, and in the group over 70 years 15.2%. These results may indicate that older members of the chamber acted with more caution. It is likely that they have reduced their workload and protected themselves more. Such behavior is only logical because there is a higher risk of fatal consequences in these age groups. Overall, the highest prevalence was recorded among women aged 30–40 and 40–50 years (32.5% and 32.4%, respectively), and the lowest among women between 60–70 years and above 70 years (19.7% and 12.1%, respectively). Additionally, a significant proportion (38.4%) stated that they were infected in the work environment.

The PCR-confirmed positivity within the population of this study was 13.9%. As of the end of this survey, the COVID-19 prevalence among the Czech general population was 15,546 cumulatively infected per 100,000 people (~15.6%) [10]. This comparison (15.6% and 13.9%) reveals that the prevalence among the respondents of this study was lower than in the general population. The difference is statistically significant (*p* = 0.0180). These outcomes suggest that although the dental profession is associated with a high occupational risk of droplet infection transmission, including SARS-CoV-19, the working conditions of dentists in the Czech Republic have not led to a higher prevalence of COVID-19 among them. Such results demonstrate that properly set working conditions focused on infection control led to a reduction in occupational infection risk.

For the majority of the respondents (97.2%), COVID-19 infection did not lead to hospitalization, and they were treated at home. However, 2.8% of the participants stated that their condition required hospitalization. This result is higher than the usual rate of COVID-19-related hospitalization. However, this may be influenced by the age composition of the respondents, as the condition for entering the chamber is a university degree in dentistry. According to Manochemi et al., the COVID-19 infection–hospitalization ratio (IHR) is 2.1% [17]. However, the IHR varies considerably across age groups, ranging from 0.4% for those younger than 40 years to 9.2% for those older than 60 years. In our study, the median age of the hospitalized individuals was 60–70 years. Among those older than 60 years, the infection–hospitalization ratio (IHR) was 8.7%. On the other hand, none of those hospitalized were under the age of 40; the IHR under the age of 40 was 0%. These findings are in accordance with those of Manochemi et al.

Overall, 37.7% of the respondents admitted that another team member was diagnosed with COVID-19, of which the most frequently mentioned profession was nurse/dental assistant (81.2%), followed by another dentist (27.4%), dental hygienist (16.7%), receptionist (12.4%), and dental technician (6.8%). These data may indicate that the distance from the site of aerosol production decreases the risk of infection. However, these results may be influenced by the uneven staffing of dental teams. Further studies would be needed to confirm this conclusion.

To compare the prevalence of COVID-19 among Czech dentists and their foreign counterparts, it is necessary to find studies of a similar methodology carried out in a similar period of time. However, a literature search revealed a lack of studies that met both of these criteria. In June 2020, a methodologically similar work was performed by the American Dental Association [15]. The questionnaire survey addressed 2195 dentists in the USA. Of them, 355 reported that they had been tested for COVID-19. Testing via respiratory, blood, and salivary samples revealed 3.7%, 2.7%, and 0% COVID-19 positivity. Despite the methodological similarity of this and our research, the data are not comparable, as they are separated by an interval of 1 year. Another online survey of dentists, dental hygienists, and dental assistants from around the world was conducted in August and September 2020 by Gluckman et al. [18]. The respondents were asked about the COVID-19 positivity among their dental practice staff. Of the total number of 1154 participants, 210 (18.2%) admitted COVID-19 infection, of which 186 (16.1%) were confirmed by a test. However, the results of this study were affected by uneven geographical participation as 48.6% of the participants were from South Africa. The COVID-19 positivity reported by the respondents from South Africa was 19%, by others 13%. Comparison with our study is, again, limited by the time difference of the event.

The literature search shows that studies focusing on the prevalence of COVID-19 among dentists are scant. Although many studies have been published focusing on the impact of COVID-19 on the operability of dental practices, current works on the impact of COVID-19 on dental professionals are lacking [13,19,20]. This condition is alarming due to the high occupational risk of dentists and emphasizes the need for further studies on this topic. Our study describing COVID-19 prevalence among members of the Czech Dental Chamber is thus one of the few that describe the impact on this professional group and, at the same time, the only one that describes this topic a year after the beginning of the pandemic.

## 6. Conclusions

This survey conducted among 2716 members of the Czech Dental Chamber reveals that 25.4% of the participants admitted to being diagnosed with COVID-19 by 30 June 2021. The total COVID-19 PCR-verified positivity was 13.9%, revealing a statistically lower prevalence (*p* = 0.0180) compared with the Czech general population (~15.6%). The results of this study suggest that although the dental profession is associated with a high occupational risk of droplet infection transmission, including SARS-CoV-19, the working conditions of dentists in the Czech Republic have not led to a higher prevalence of COVID-19 infection among them. Such results demonstrate that properly set working conditions focused on infection control were effective and led to a reduction in the occupational infection risk.

## Figures and Tables

**Figure 1 ijerph-18-12488-f001:**
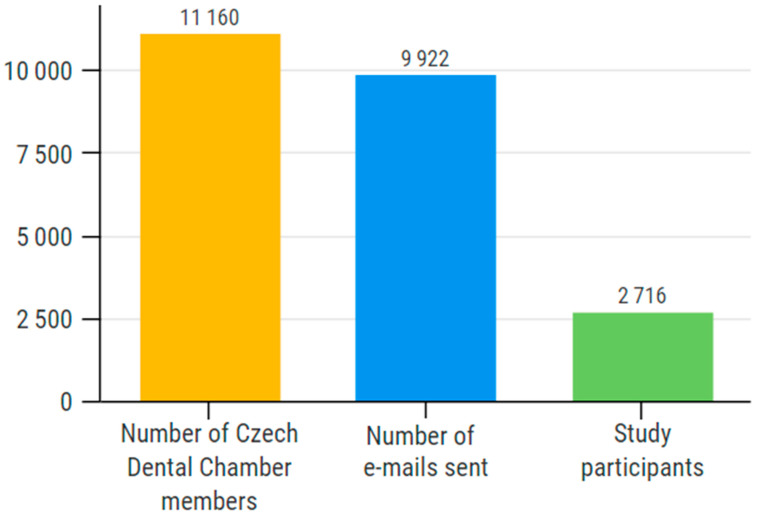
Response rate. The 2716 participants represent 27.4% of all the e-mail addresses included and 24.3% of the Czech Dental Chamber members.

**Figure 2 ijerph-18-12488-f002:**
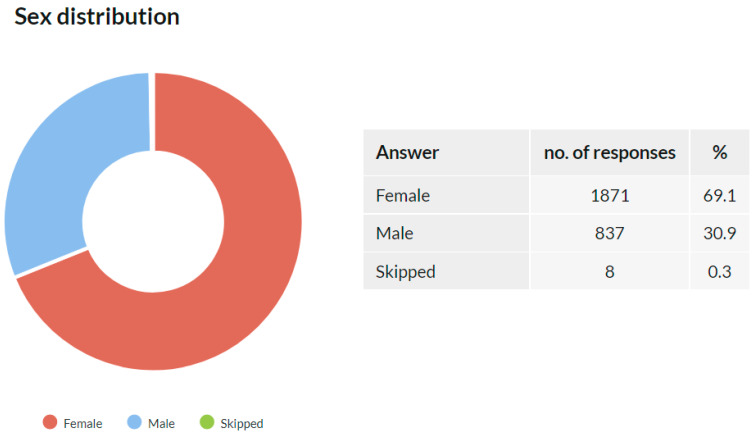
Sex distribution of the study participants.

**Figure 3 ijerph-18-12488-f003:**
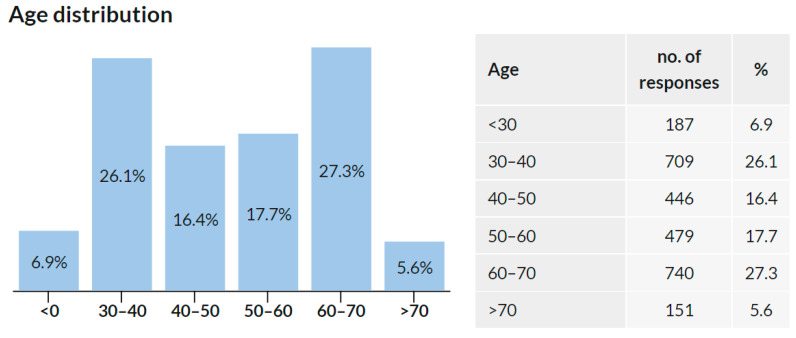
Age distribution of the study participants.

**Figure 4 ijerph-18-12488-f004:**
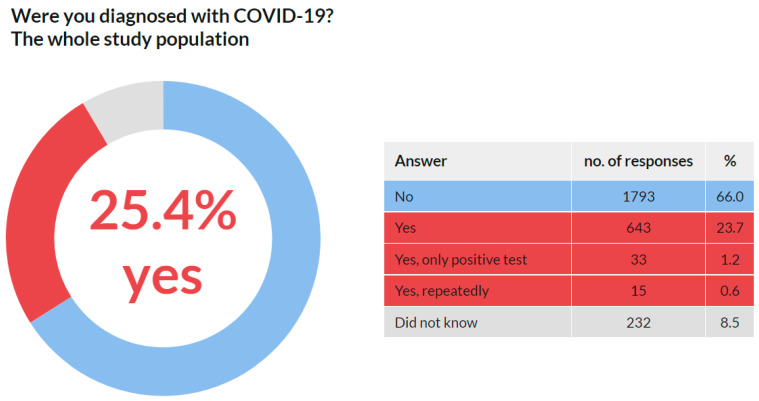
COVID-19 positivity, the whole study population.

**Figure 5 ijerph-18-12488-f005:**
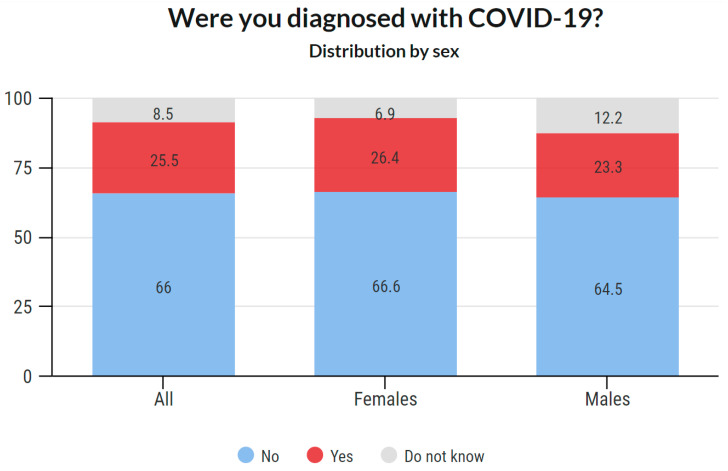
COVID-19 positivity, distribution by sex.

**Figure 6 ijerph-18-12488-f006:**
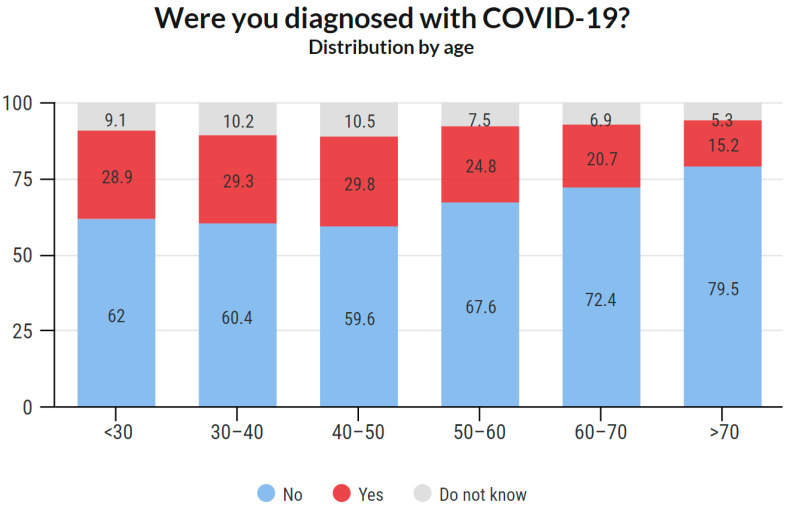
COVID-19 positivity, age distribution.

**Figure 7 ijerph-18-12488-f007:**
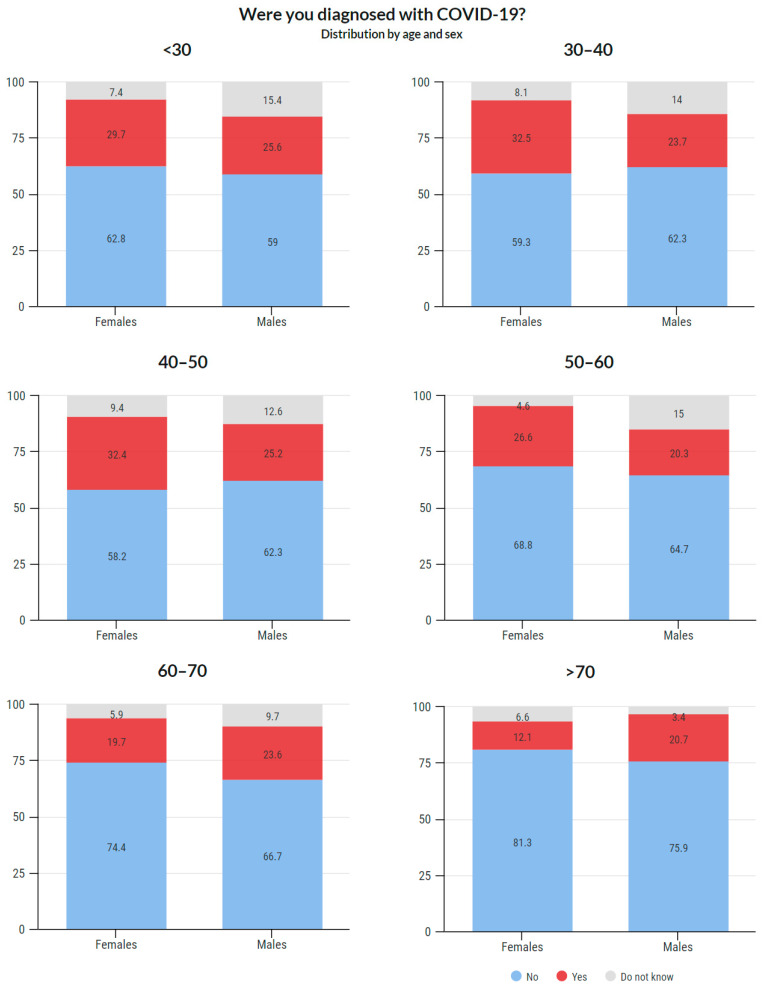
COVID-19 positivity, age and sex distribution.

**Figure 8 ijerph-18-12488-f008:**
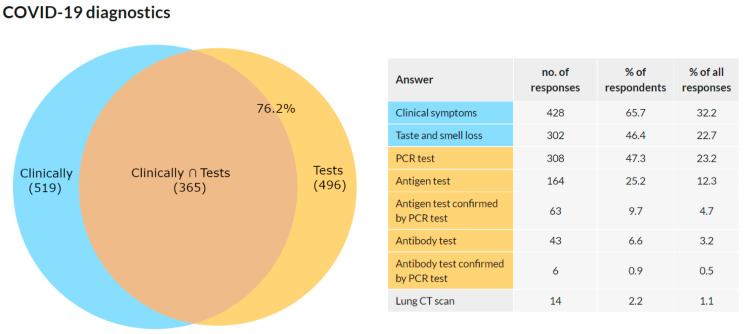
COVID-19 diagnostics.

**Figure 9 ijerph-18-12488-f009:**
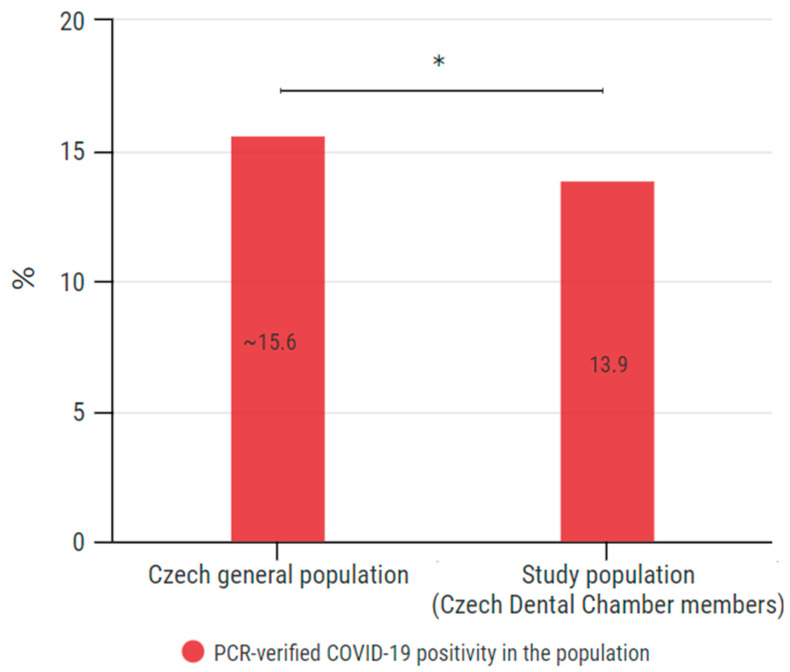
Comparison of COVID-19 prevalence in the Czech general population and its estimation within the population of Czech Dental Chamber members. Chi-square with test Yates’s correction was used for statistical analysis; *p* = 0.0180. * indicates *p* < 0.05.

**Figure 10 ijerph-18-12488-f010:**
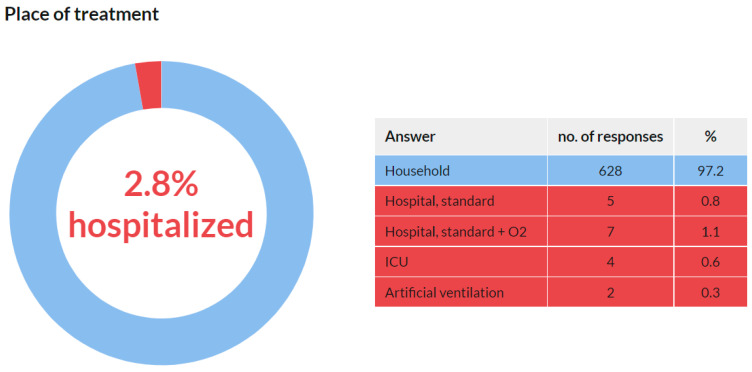
Place of treatment.

**Figure 11 ijerph-18-12488-f011:**
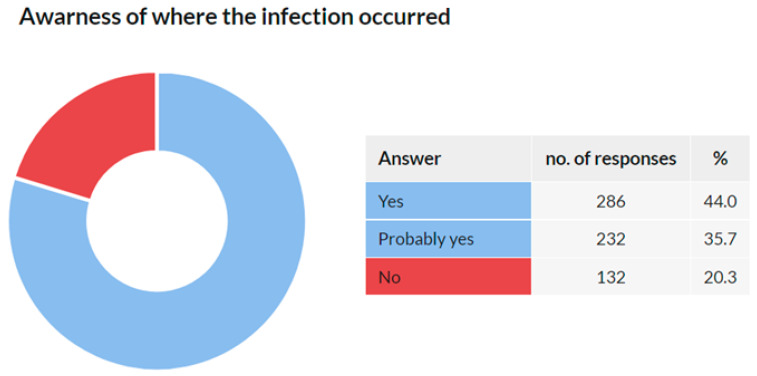
Awareness of where the infection occurred.

**Figure 12 ijerph-18-12488-f012:**
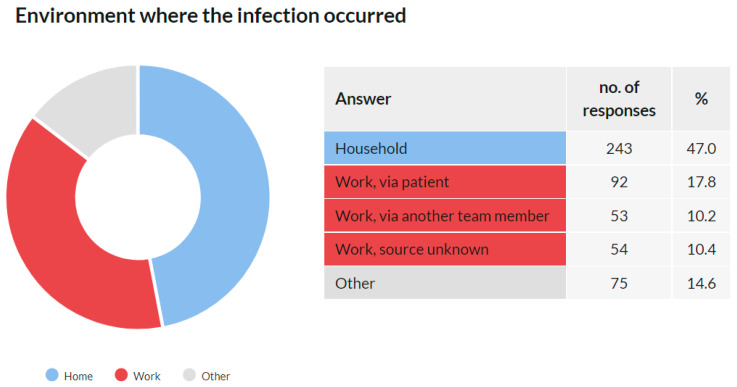
The environment where the infection occurred.

**Figure 13 ijerph-18-12488-f013:**
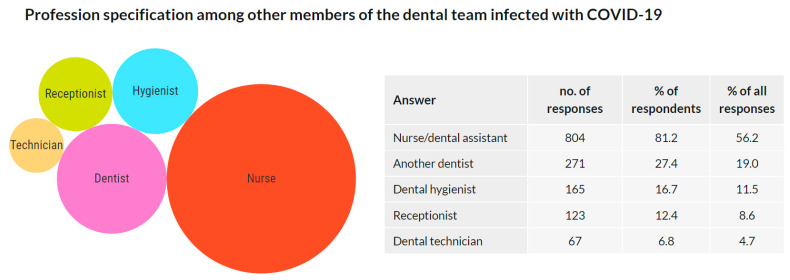
Profession specification among other members of the dental team infected with COVID-19.

**Table 1 ijerph-18-12488-t001:** Questions and their classification.

Question Mark	Question	Question Type	Number of Closed-Ended Answer Options	Answer Choice
Q1	Sex	Closed	2	Single
Q2	Age	Closed	6	Single
Q3	Were you diagnosed with COVID-19 by 30 June 2021?	Closed	3	Single
Q4	How was COVID-19 diagnosed?	Closed	8	Multiple
Q5	Where did the treatment take place?	Closed	5	Single
Q6	Do you know where you got infected?	Closed	3	Single
Q7	Where did the transmission of COVID-19 occur?	Semiclosed	6	Single
Q8	Was another member of the team diagnosed with COVID-19?	Closed	2	Single
Q9	Which team member was it?	Closed	5	Multiple

## Data Availability

The dataset is available on demand from the corresponding author.

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
