# Peer review of "COVID-19 Prevalence among Czech Dentists"

_ijerph, 2021, doi:10.3390/ijerph182312488_

Round 1

Reviewer 1 Report

This study presents important epidemiological data and is therefore of high relevance.

The Introduction and Methods section are well presented. The results section is also well presented regarding the COVID-19 prevalence among dentists. However, there is a major flaw in the presentation of  the other results and the conclusions drawn from it.

The authors state that the prevalence of COVID-19 among dental assistants is 81.2 % (!!!), see Fig. 13. If true, that would be a devastating result that would have to lead to the conclusion that anti-epidemic measures were NOT efficient. One would also have to seriously consider how such a significant difference between COVID-19 infections of dental assistants and dentists could have taken place. Were protective masks only handed to dentists?!

It is unclear from the data whether the dentists also provided information on how many dental assistants they had working with them. I therefore think it probably that in 80% of cases, at least one assistant was infected, leading to this high number. Then, however, this percentage is not the actual prevalence of COVID-19 among the dental assistant profession, as there are usually many more assistants than dentists in one practice.

Whatever the case, this information must be clarified and presented in a  way that avoids misunderstandings regarding the prevalence of COVID-19 infections among different professions in the field of dentistry.

Reviewer 2 Report

This is an interesting and relevant paper on the nationwide prevalence of Covid-19 disease among dental practitioners. Czech dentists are indeed an interesting population to study given that they mostly maintained their work during the lockdown period. I find the investigation well-performed and the manuscript logically and clearly written. However, I would like to ask the authors to reduce unnecessary repetition, which occurs quite a lot. There are parts of the Introduction section and Materials and methods section that are being extensively repeated in the Discussion; this is largely unnecessary. Please condense the paper to present the important information only once in the section where they belong, instead of repeating them multiple times. The paper could be approximately 30% shorter if all the repetitions were excluded. Also, in the results section, please write only about the data you wish to highlight, it is unnecessary to mention in the text all numerical data that has also been shown in figures/tables. This is especially pronounced in the demographics section, where the Results section repeats practically everything that is already shown in figures/tables. In summary, please try to condense and make the manuscript more concise, the readers should appreciate that.

Round 2

Reviewer 1 Report

Many thanks for your revisions, I believe the manuscript can now be accepted for publication in its current form.

Reviewer 2 Report

Dear authors,

Thank you for submitting the revised version of this interesting manuscript. I have no further remarks at this point. The paper can be accepted for publication.